# Inhibitors of ATP Synthase as New Antibacterial Candidates

**DOI:** 10.3390/antibiotics12040650

**Published:** 2023-03-24

**Authors:** Rawan Mackieh, Nadia Al-Bakkar, Milena Kfoury, Rabih Roufayel, Jean-Marc Sabatier, Ziad Fajloun

**Affiliations:** 1Faculty of Sciences 3, Department of Biology, Lebanese University, Campus Michel Slayman Ras Maska, Tripoli 1352, Lebanon; 2Faculty of Health Sciences, Beirut Arab University, Beirut Campus, Riad El Solh, Beirut 1105, Lebanon; 3College of Engineering and Technology, American University of the Middle East, Egaila 54200, Kuwait; 4CNRS, Institute of Neurophysiopathology, Aix-Marseille Université, 13385 Marseille, France; 5Laboratory of Applied Biotechnology (LBA3B), Azm Center for Research in Biotechnology and Its Applications, EDST, Lebanese University, Tripoli 1300, Lebanon

**Keywords:** ATP synthase, ATP synthase inhibitors, therapeutic application, animal venoms, resistant bacteria

## Abstract

ATP, the power of all cellular functions, is constantly used and produced by cells. The enzyme called ATP synthase is the energy factory in all cells, which produces ATP by adding inorganic phosphate (Pi) to ADP. It is found in the inner, thylakoid and plasma membranes of mitochondria, chloroplasts and bacteria, respectively. Bacterial ATP synthases have been the subject of multiple studies for decades, since they can be genetically manipulated. With the emergence of antibiotic resistance, many combinations of antibiotics with other compounds that enhance the effect of these antibiotics have been proposed as approaches to limit the spread of antibiotic-resistant bacteria. ATP synthase inhibitors, such as resveratrol, venturicidin A, bedaquiline, tomatidine, piceatannol, oligomycin A and N,N-dicyclohexylcarbodiimide were the starting point of these combinations. However, each of these inhibitors target ATP synthase differently, and their co-administration with antibiotics increases the susceptibility of pathogenic bacteria. After a brief description of the structure and function of ATP synthase, we aim in this review to highlight therapeutic applications of the major bacterial ATP synthase inhibitors, including animal’s venoms, and to emphasize their importance in decreasing the activity of this enzyme and subsequently eradicating resistant bacteria as ATP synthase is their source of energy.

## 1. Introduction

Life requires energy. The universal, biological fuel is a molecule called ATP or adenosine triphosphate. ATP stores chemical energy in the form of a high-energy phosphate bond. The most frequent chemical process in biological reactions is ATP production. The very last enzyme in the oxidative phosphorylation pathway to employ electrochemical energy to trigger ATP synthesis is called ATP synthase (also known as F-ATPase or F0F1-ATP). This enzyme produces ATP from its precursor, ADP and inorganic phosphate (Pi) [1,2,3,4]. ATP synthases are key components of cellular bioenergetics and are found in all living things on earth: the cristae and inner membrane of mitochondria, the thylakoid membrane of chloroplasts, and the plasma membrane of bacteria [5]. Additionally, F-ATPases are universal in their overall structure and function as rotary energy converters. Bacterial ATP synthases have a simpler subunit structure than their mitochondrial counterparts. However, the mitochondrial genome of eukaryotes encodes subunit α, which restricts genetic analysis of the functions of various residues. Contrarily, subunit α and β of bacteria have been the subject of a great deal of mutagenesis research, with *Escherichia coli* ATP synthase being the most commonly investigated [6]. On the other hand, antimicrobial resistance puts several aspects of contemporary medicine at risk, including the recurrence of potentially fatal bacterial diseases. No new classes of antibiotics have been produced into new treatments for decades, despite rigorous drug discovery efforts. This is mostly because effective antibiotic drugs must meet strict chemical, biological and pharmacological requirements. Therefore, studies have focused on finding radical solutions to fight the challenging resistant bacteria, a real complication that is increasing day by day. Thus, ATP synthase was the center of attention due to its critical role in bacterial survival.

In this review, we discussed the importance of inhibiting the ATP synthase in bacteria in order to find an alternative to antibiotics to face resistant bacteria and to illustrate the importance of this enzyme in keeping the bacteria alive which makes it a real therapeutic target.

## 2. Structure and Function of ATP Synthase

The exceptional function of ATP synthase lies in its remarkable structure that offers it the ability of working in a particular way. ATP synthase comprises an assembly of two rotary motors F1 and F0. F1 (~380 kDa) is water soluble and possesses catalytic sites for ATP synthesis/hydrolysis [7]. It is known as F1-ATPase because, when separated from the membrane section, it functions as an ATP-driven motor that rotates its inner subunit to hydrolyze ATP [8]. The bacterial F1 sector is located at the exterior side of the membrane and is formed by an arrangement of catalytic α3β3 hexamer stator rings in which subunits α and β are alternately organized (Figure 1).

Those subunits are accompanied by subunit γ (rotor shaft) in the space based at the center of the ring. Additionally, there are subunits δ and ε that are hitched to α and γ consecutively [8,9,10]. The subunit δ serves as a connection to join the stator components between F1 and F0. The ε one acts as a connection between the rotor parts of F1 and F0, it is also considered as an endogenous inhibitor of F1 by causing conformational changes that switch the state from closed to extended form, resulting in a subunit γ rotation blockade as a result of steric hindrance [8,11,12]. It is believed that the physiological importance of this inhibitory action is to prevent ATP consumption [13]. On another hand, the second motor F0 (~120 kDa) is incorporated in the membrane and constitutes a proton route created by the assembly of subunits ab2c10-15 [14,15]. An aligned circle is formed by subunit c that constitutes a ring complex [8]. Every bacterial species has a different number of subunit c ; for instance, the c component has 10 copies in *E. coli* [16] and thermophilic *Bacillus* PS3 [17], 11 copies in *Ilyobacter tartaricus* [18,19] and *Clostridium paradoxum* [20], 13 in thermoalkaliphilic *Bacillus* TA2.TA1 [21] and *Bacillus pseudofirmus* OF4 [22], and 8 copies in bovine mitochondria [23] (Table 1). Concerning the difference between the mitochondrial and bacterial structure of ATase; both of them possess the same core of subunits, but the mitochondrial one has additional subunits in the F0 domain which are subunit DAPIT (diabetes-associated protein in insulin-sensitive tissues) and 6.8 kDa proteolipid, and it also contains subunits d and F6 in the stator part. Moreover, there is OSCP (Oligomycin sensitivity-conferring protein), which is similar to subunit δ of the bacterial enzyme [24].

The c-ring rotates in the reverse direction to subunit γ of the F1 motor against subunits ab2 during the downhill proton flow through the proton channel [25]. As a result, in the F0F1 complex, F0 and F1 force each other in opposite directions. Contrariwise, F0 forcibly rotates subunit γ clockwise under physiological conditions where the electrochemical potential of the protons is high enough to exceed the free energy of ATP hydrolysis. F1 then catalyzes the reverse reaction, which is ATP synthesis, which is the primary physiological function of ATP synthase. F0 is forced by F1 to rotate the c-ring in the opposite direction in order to pump protons against the electrochemical potential when the electrochemical potential is low or reduced [8]. It is notable that besides producing ATP, and when the electrochemical potential is insufficient, this enzyme can also work in a reversed way by catalyzing proton pumping, which creates the electrochemical potential needed to hydrolyze ATP into ADP and Pi [8]. This makes from ATP synthase a ubiquitous enzyme (Figure 2).

Similarly to the bacterial ATPase, the mitochondrial enzyme consists of two domains, F1 and F0. The F1 domain is located outside the mitochondrial membrane and contains the catalytic site for ATP synthesis, while the F0 domain resides in the membrane. The F1 domain of the mitochondrial enzyme is also composed of subunits α, β, γ, ε and δ; however, the F0 domain is characterized by containing a different number of subunit c copies (eight in bovine mitochondria) and other minor subunits such as A6L, e, f and g conferring a higher level of complexity to the mitochondrial ATPase (Figure 1).

The stoichiometry of the globular F1 subunits is as follows: α3β3γ1δ1ε1. subunits A and B surround antiparallel α helices in the subunit γ, which rotates when protons pass through the F0 domain. Catalytic sites can be found at the interface of subunit α/β , and δ and ε interact with γ in the central stalk, which contains subunit OSCP [23,26,27].

## 3. Therapeutical Applications

Since bacterial ATP synthase has emerged as a prospective therapeutic target to develop a new strategy to treat infections brought on by resistant bacteria, many studies have been conducted to produce different combinations between antibiotics and bacterial ATP synthase inhibitors. These combination therapy approaches are rapidly being studied to prolong the effectiveness of currently available antibacterial medications. Thus, different types of bacterial ATP synthase inhibitors may be used as promising antibacterial therapy.

### 3.1. Chemical Inhibitors

#### 3.1.1. Resveratrol

A previous study has shown that effectiveness of aminoglycosides against *Staphylococcus aureus* is improved by resveratrol, a polyphenolic ATP synthase inhibitor that is usually given as a dietary supplement [28]. This inhibitor is known to bind between subunit β and the c-terminal region of subunit γ [29]. Additionally, during therapy with aminoglycosides, resveratrol prevents the development of de novo resistance [28]. In addition, deactivating ATP synthase by resveratrol makes *S. aureus* 16-fold more susceptible to the aminoglycoside, gentamicin [28]. Moreover, the application of resveratrol on *S. aureus* makes this pathogen more susceptible to hBD4-mediated killing (antimicrobial peptides of the innate immune system) [30]. Interestingly, resveratrol does not seem to have harmful or incapacitating adverse effects, and this is confirmed by long-term clinical trials [31,32].

#### 3.1.2. Piceatannol

Piceatannol (Figure 3a), a remarkable polyphenolic inhibitor similar to resveratrol, has shown a potential inhibition of ATP synthase by interacting with the pocket created by contributions from subunits α and β stator and the carboxyl-terminal region of subunit γ rotor [33]. Piceatannol was shown to have the strongest inhibitory impact on *E. coli*, in comparison with other polyphenolic inhibitors such as resveratrol, quercetin, quercitrin and quercetin-3-β-D glucoside, with almost total inhibition of ATP synthase (approximately 0 residual activity) [34]. Additionally, it is important to mention that both piceatannol and resveratrol had inhibited ATP synthase activity and ATP synthesis, while the other studied inhibitors prohibited only ATP synthase activity [34]. On the other hand, normally, the ATP synthase F1 sector rotates continuously through cycles of catalytic dwells (~0.2 ms) and 120° rotation steps (~0.6 ms). However, piceatannol prolonged the length of the catalytic dwell but did not impede movement during the 120° rotation stage [33]. Using piceatannol also resulted in a notably longer duration of the intrinsically inhibited state of F1 [33]. Additionally, piceatannol, with other inhibitors such as curcumin and desmethoxycurcumin, has established an inhibitory effect on *Streptococcus mutans,* a significant pathogen of dental caries which primarily lives on the surface of teeth. The *S. mutans* becomes very susceptible to F-ATPase inhibitors in acidic environments, knowing that this pathogen crucially required F-ATPase for its ability to tolerate acidity [35]. The use of piceatannol does not harm the human health, but in contrast, it has a potential therapeutic effect such as the prevention of cardiovascular diseases [36,37].

#### 3.1.3. Bedaquiline

Bedaquiline (Figure 3b) is the first ATP synthase inhibitor against multidrug-resistant tuberculosis. According to World Health Organization (WHO), tuberculosis (TB) is the ninth most common infectious disease-related cause of death in the world [38]. Countless studies have been conducted into finding the most efficient treatment toward TB, particularly to tackle concurrently increasing drug-resistant mycobacterial strains. One of these treatments is based on bedaquiline, also known as diarylquinoline TMC 207. This drug has become a mainstay of therapy for tuberculosis because it can eradicate even latent *Mycobacterium tuberculosis* infections without causing toxic effects on ATP production in mammalian cells [39,40,41,42,43]. Previous studies have shown that bedaquiline targets mycobacterial ATP synthase by attaching to a specific binding site on subunit c, thereby impeding the rotational movement of this subunit during catalysis or, alternatively, at the interface between the oligomeric subunit c and subunit a [44,45]. This substance’s exceptional ability to effectively eradicate mycobacteria in a variety of microenvironments may be partially explained by the strong affinity of TMC207 at low pH and low proton motive force values [44]. Furthermore, it has been reported that bedaquiline can bind to ε subunit [46,47]. Andries et al. have demonstrated that bedaquiline has a potent inhibitory effect even against resistant *M. tuberculosis*, and the substitution of the typical TB treatment (rifampin, isoniazid, and pyrazinamide) by bedaquiline has enhanced the bactericidal activity [48]. It is important to mention that bedaquiline has a selective effect which means it works against mycobacteria but shows limited or no direct antibacterial effect against other pathogenic bacteria, such as Gram-positive *Nocardia*, *Corynebacterium*, *Streptococcus pneumoniae*, *S. aureus* and *Enterococcus faecalis*, or Gram-negative *E. coli*, *Helicobacter pylori*, and *Haemophilus influenzae* [48]. Researchers were able to enhance the inhibitory activity by combining bedaquiline with other antibacterial compounds. A human embryonic stem cell reporter assay coupling GaMF1, a novel antimycobacterial, with bedaquiline or new diarylquinoline compounds resulted in potentiation without causing genotoxicity or phenotypic alterations [49]. However, faster death of *M. tuberculosis* is made possible by combining bedaquiline and substrate-level phosphorylation in glycolysis/gluconeogenesis [50]. The fact that makes bedaquiline an interesting drug is its remarkable effect on the host immune system. It has been discovered in a recent study that bedaquiline has enhanced host macrophage innate immune resistance to bacterial infection [51]. However, according to early studies where macrophages have been cultured with and without bedaquiline, in both cases, macrophages have developed normally [51]. It is true that bedaquiline has a very potential inhibitory effect, but some studies have shown that it presents at some point adverse effects that could be life-threatening [52]. Additionally, a previous study has demonstrated that bedaquiline may inhibit the mitochondrial ATP synthase [53], in addition to interfering with other drugs included in TB treatment [52]. Due to the important effect of bedaquiline, the management of these adverse effects is much required so it is suggested to produce derivatives of bedaquiline to increase its specificity toward the enzyme of *M. tuberculosis* and to reduce the harmful effect on human health [53].

#### 3.1.4. Tomatidine

Tomatidine (TO) (Figure 3c), a notable inhibitor and steroid alkaloid [54], possesses a strong bactericidal activity by targeting subunit c of ATP synthase [55]. Tomatidine exhibits a bacteriostatic activity (MIC of 0.12 μg/mL) toward *S. aureus* small colony variants (SCVs) that are linked to infections caused by this bacteria, including those that affect people with cystic fibrosis (CF) [56]. These findings indicate that tomatidine may eventually be used in combination therapy with other conventional antibiotics to get rid of *S. aureus* strains that are tenacious, which is confirmed by many studies. TO was proved to be an enhancer of aminoglycosides that also functions as an antivirulence agent, emphasizing on both antibiotic-susceptible and antibiotic-resistant *S. aureus* [54,57]. Additionally, it has been demonstrated that TO induces the ROS production which increase the gentamycin cell uptake against *S. aureus* prototype, confirming that TO and the aminoglycosides can work in synergy [58]. When treating infections caused by *S. aureus*, *Pseudomonas aeruginosa*, and *E. faecalis*, tomatidine may be used as an antibiotic potentiator in combination with gentamicin, cefepime and ciprofloxacin, as well as ampicillin [59]. TO exhibits a potent bactericidal action against *S. aureus* when cocultured with *P. aeruginosa*. As a result, for individuals with cystic fibrosis who are commonly co-colonized by MRSA and *P. aeruginosa*, the combination of TO and tobramycin may provide a new therapeutic strategy [60]. Moreover, the bactericidal activity of TO against *Listeria monocytogenes* and its antibacterial activity against prototype *S. aureus* were both improved by chemical changes. Antibacterial properties against such well-known organisms may be helpful in treating MRSA infections or preventing *Listeria* contamination of the food chain [61].

#### 3.1.5. N,N-dicyclohexylcarbodiimide

Another well-known inhibitor of the F0F1-ATP synthase is N,N-dicyclohexylcarbodiimide (DCCD), which forms a covalent bond with the remarkably conserved carboxylic acid of the proteolipid subunit (subunit c) in F0 [62]. It is known that DCCD has the power to inhibit *E. coli* [62,63].

A recent study compared the inhibitory effect between many ATP synthase inhibitors, including DCCD, tomatidine, resveratrol and piceatannol on *S. aureus* small colony variants and *Streptococcus pyogenes*. The results showed that, except for *S. aureus*, which does not require ATP synthase for growth, DCCD exhibited broad-spectrum inhibitory efficacy against all strains with a minimum inhibitory concentration (MIC) of 2–16 g/mL. However, when tested against *S. aureus* small colony variations with reduced electron transport chain activity, tomatidine displayed extremely powerful yet selective action (MIC 0.0625 g/mL). Piceatannol inhibited *S. pyogenes* at MIC 16–32 g/mL, and *S. aureus* small colony mutants were also more sensitive to resveratrol and piceatannol than the wild-type strain [64].

### 3.2. Inhibitors Isolated from Bacteria

Some compounds isolated from different bacteria can act as ATP synthase inhibitors and thus be used in therapeutic applications.

#### 3.2.1. Oligomycin A

Oligomycin A is an inhibitor usually used because of its effect toward ATP synthase, which binds to c-ring proton carrying sites (Figure 4) [65] of yeast, but previous genetic studies contend that the oligomycin A-binding site forms a common “drug binding site” with the binding sites of other antibiotics, such as those that are effective against *M. tuberculosis*. Therefore, it is suggested that new antibiotics can be created through rational design which effectively target this drug-binding region [26]. However, oligomycin A is non-selective and blocks ATP synthase in mitochondria, which limits its clinical use [39].

#### 3.2.2. Venturicidin A

A naturally occurring substance called venturicidin A, a specific F0 directed inhibitor isolated from *Actinomycetes* that targets the subunit c, has significantly altered the affinity with suppression of both ATP production and hydrolysis, additionally served as a novel adjuvant for aminoglycosides and recovers gentamicin’s effectiveness against drug-resistant clinical isolates [66,67,68,69]. Unfortunately, similarly to oligomycin A, venturicidine A inhibits both the mitochondrial and bacterial ATP synthase [70,71]. Although there are many approaches that could lower the specificity of venturicidin toward mitochondrial ATP synthase, such as generating a better analog that exhibits stronger selectivity toward bacteria by identifying the biosynthetic gene of cluster venturicidine A in the producer strain and performing logical biosynthetic alterations [69].

### 3.3. Natural Inhibitors

Besides different molecules mentioned before in this review, there are natural compounds that exhibit a level of inhibition against the bacterial ATP synthase enzyme, and are worthy to be studied further for potential application.

#### 3.3.1. Natural Spices

Ginger and curcumin, known for their antioxidant, anticancer and antibacterial effects, can also exhibit a strong inhibitory effect on ATP synthase. Curcumin, a dietary phyto-polyphenol derived from the perennial herb curcuma longa, inhibits F1-ATP synthase activity by interfering with conformational changes at subunit β catalytic region and prolonging the catalytic dwell [72,73]. Additionally, the inhibitory mechanism of curcumin differs from other polyphenols [73]. In the same manner, dietary ginger phenolics (DGPs) have completely (100%) inhibited the wild-type *E. coli* F1F0-ATP synthase that was attached to the membrane [74]. Ginger and curcumin have many medicinal uses and show many other benefits on human health, but further studies are required concerning the dosage and medication frequency [75,76].

Table 2 summarizes most of the bacterial ATP synthase inhibitors available to date, to the best of our knowledge, by presenting the different targeted subunits of ATP synthase in selective bacteria.

#### 3.3.2. Animal Venoms

Considering the disadvantages of synthetic antimicrobial agents (cost, resistance, side effects), researchers in many health fields tend to include natural agents in their studies concerning treating infections. Common natural agents used, in addition to the inhibitors listed and discussed before, are animal venoms (bees, ants, scorpions, spiders, snakes, etc.). Venoms produced by animals have been used for several years in the creation of novel medications to treat conditions such as cancer, hypertension and inflammation, as well as analgesic medications [84]. A previous study has demonstrated that *Apis mellifera* bee venom (BV-Am) has potentially inhibited *E. coli* F0F1-ATP synthase, proving that venoms are effective against many bacterial infections by targeting the ATP synthase. It is known that *Apis mellifera* venom contains two active components, melittin and phospholipase A2 (PLA2), which are responsible for the venom’s activity [77]. Melittin obstructs the ATP synthase activity of F1 [78]. The *Montivipera bornmuelleri* snake venom has also demonstrated an inhibitory effect due to two active components: PLA2 and L-amino-acid-oxidase (LAAO). This venom was able to inhibit the ATP synthase of Gram-positive *Staphylococcus epidermidis* and Gram-negative *E.coli* with a concentration of the order of 100–150 µg/mL, offering a clear advantage: it can target a wide spectrum of bacteria [81]. On the other hand, many venom peptides have been studied, such as anoplin, cupiennin 1a, latarcin 1, latarcin 3a, latarcin 5, and pandinin 2, and have revealed an inhibitory capacity by targeting the βDELSEED-motif of ATP synthase [79]. Another study has shown that introducing some modifications on c-terminal-NH2 groups of some insects’ venoms (eumenitin, lasiocepsin, lycosin1, mastoparanB, panurgine1 and protonectin) has amplified the inhibitory effect on ATP synthase by targeting the βDELSEED-motif in this enzyme [82]. Similarly, another study has confirmed the importance of the presence of c-terminal-NH2 groups in venom peptides, and adding this group has increased the inhibitory effect 100-fold [85]. The motif is surrounded by more than twenty variable amino acids, although, even among bacteria, this area is not conserved in terms of length or amino acid composition [13,85]. It is noteworthy to mention that the βDELSEED-motif is highly conserved all over evolution, which makes from it a special target to take advantage of and use to eradicate bacteria. Likewise, king kobra, banded krait and wolf spider venom peptides (cathelicidin KF-34, cathelicidin BF-30 and lycotoxin I-II, respectively) have shown a great inhibitory activity ≥85% against wild-type *E. coli* [83]. Furthermore, venom peptide cathelicidin BF-30 was shown to be effective against drug-resistant *E. coli*, *P. aeruginosa* and *S. aureus* [83]. Nevertheless, it is true that these toxins proved to have an inhibitory effect against bacterial ATP synthase, but clinical studies have to be conducted concerning the intake dosage and to verify if the mitochondrial ATP synthase is affected [86].

From another point of view, genetic modifications of the bacterial ATP synthase may inhibit its function and thus serve as a future tool for antimicrobial treatment. In fact, several genetic manipulations have been performed on the bacterial genome to impact the function of the ATP synthase. Hotra et al. demonstrated that the deletion of the relevant loop-encoding sequence (166–179) from the *Mycobacterium smegmatis* genome resulted in greater ATP cleavage and reduced levels of ATP synthesis when compared to the wild-type enzyme, proving that the loop influences ATP synthase activity, ATP synthase driven H(+) pumping and ATP synthesis [87]. Another study demonstrated that producing mutations in the ATP synthase makes *S. aureus* more susceptible to neutrophils than the wild type [30]. Additionally, mutating *atpA* gene in the target bacteria significantly enhanced the inhibitory potential of AMPs, which are antimicrobial peptides responsible for attacking bacterial pathogens. In comparison to wild type cells, ATP synthase mutants (mutation in *atpA* gene) of *S. aureus* become more sensitive to death by AMPs such as β-defensins (hBD4 and hBD2), LL-37 and histatin 5 [30].

However, it is significant to note that human ATP synthase differs greatly from bacterial ATP synthase, which is a promising approach that facilitates the development of new ATP synthase inhibitors that are safe to be used in humans [39].

Finally, in this review, we focused on ATP synthase inhibitors as new candidates to face resistant bacteria. However, the possibility of the emergence of ATP synthase inhibitor-resistant bacteria should always be expected since in previous studies, researchers have found strains of *E. coli* that are resistant to DCCD [88,89]. In fact, if there is not good management and awareness concerning the use of previous and new antibiotics and other antibacterial agents, we will eventually reach the same point, which is resistant bacteria, because as we know, bacteria have high mutational ability to adapt, which makes the possibility of resistance higher. However, only a comprehensive and integrated set of measures involving numerous accountable parties will be able to control this issue. We recommend concrete steps that the responsible instances can take to optimize antibiotic stewardship and revive antibiotic development. Additionally, it is important to impose restriction on using antibiotics to encourage animal growth.

An interesting new target to combat the emergence of antibiotic-resistant bacteria is the TrwD ATPase, an enzyme that powers different steps of bacterial conjugation, a mechanism used to transfer antibiotic resistance genes to new bacteria. Using synthetic fatty acids, the inhibition targeted only bacterial conjugation, resulting in a lack of cell death. This fact could be useful to keep commensal bacteria alive, promoting gut health. Further research should be poured into this area, for a new approach to antibacterial resistance and a more potent result [90].

Henceforth, the ability to inhibit only the bacterial ATP synthase is a major key point in the creation of new inhibitors that will not interfere with the host cell enzyme, avoiding the disadvantage of a substance with a broader area of action; an inhibitor which would target both prokaryotic and eukaryotic ATP synthase would kill both the bacteria and the host cell, which is a wholly undesired side effect.

Figure 4 illustrates the different bacterial ATP synthase inhibitors described in this review, with their sites of action on this enzyme.

## 4. Conclusions

Ultimately, the ATP synthase plays an important role in energy metabolism as a ubiquitous enzyme. The difference between bacterial and human ATP synthase is a strategic opportunity to develop novel antibacterial treatments without affecting human health. Additionally, with an ever rising need for novel antibiotics due to the phenomenon of antibiotic resistance, serious research into creative workarounds for this problem is highly encouraged, and as this review made clear, one of the promising therapies is the ATP synthase inhibitors. Many ATP synthase inhibitors have shown a prominent and selective effect against certain bacterial species. Their use offers an unprecedented efficacy and selectivity, as well as other advantages, such as bedaquiline’s enhancement of the host immune system response to infection. This ability merits further investigation because it could have a huge beneficial effect on cancer patients whose immune system is weak. Emphasis should be placed on counteracting the lack of specificity of certain inhibitors such as oligomycin A and venturicidin A, and on the large concentration of natural products used to obtain sufficient efficacy. The rational design of novel, selective inhibitors of bacterial ATP synthases may someday benefit from improved understanding of the activity of this important enzyme across species.

## Figures and Tables

**Figure 1 antibiotics-12-00650-f001:**
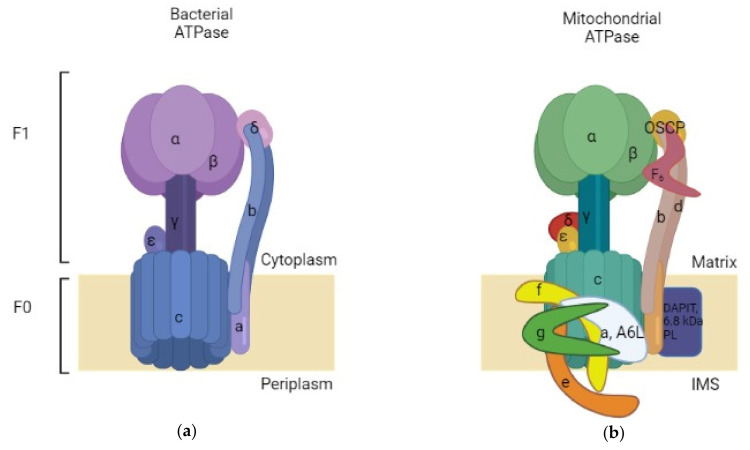
Simplified comparison between bacterial and mitochondrial ATPases. (**a**) Bacterial ATPase: comprised of two subunits: F0, inserted in the membrane, and F1, which is present in the bacterial cytoplasm; (**b**) Bovine mitochondrial ATPase: also comprised of a membrane-inserted F0 and an F1 subunit situated in the mitochondrial matrix. The mitochondrial ATPase is more complex than the bacterial enzyme. Made with BioRender.

**Figure 2 antibiotics-12-00650-f002:**
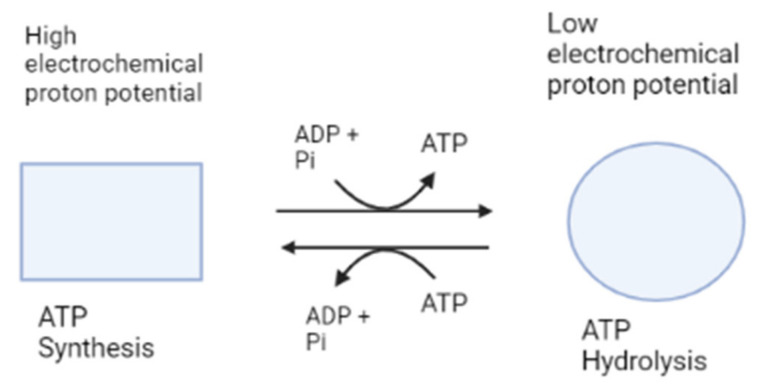
Simplified ATP synthase mechanism of action. This figure shows the two functions of ATP synthase, represented as different shapes, synthesis and hydrolysis of ATP. Made with BioRender.

**Figure 3 antibiotics-12-00650-f003:**
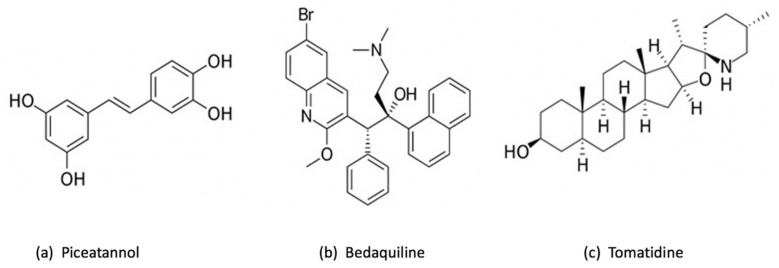
Chemical structure of 3 chemical inhibitors: (**a**) Piceatannol, (**b**) Bedaquiline and (**c**) Tomatidine. Made with BioRender.

**Figure 4 antibiotics-12-00650-f004:**
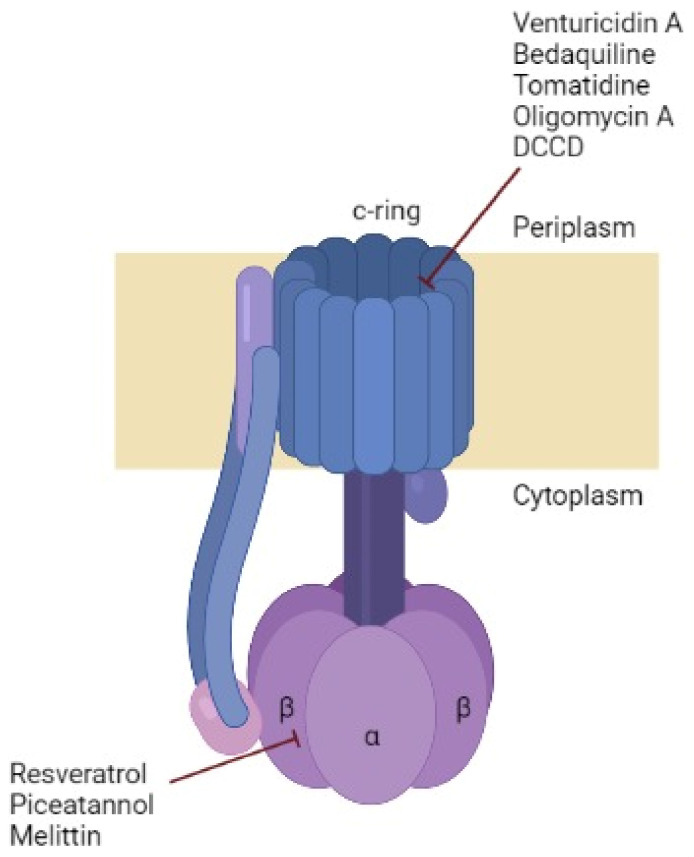
Bacterial ATP synthase composition and the subunit targeted by some inhibitors. ATP synthase represents the typical structure identified in bacteria. The number of monomers in c-ring differs between organisms. The inhibitory sites of each inhibitor are showed in red. Made with BioRender.

**Table 1 antibiotics-12-00650-t001:** Number of subunit c copies in ATPase of different species. This table represents bacterial and eukaryotic species and the disparity between the number of subunit c copies of their ATPase enzymes.

Species	Number of Subunit c Copies
*E. coli*	10
Thermophilic *Bacillus PS3*	10
*I. tartaricus*	11
*C. paradoxum*	11
Thermoalkaliphilic *Bacillus TA2*, *TA1*	13
*B. pseudofirmus OF4*	13
*Bovine Mitochondria*	8

**Table 2 antibiotics-12-00650-t002:** Summary of all ATP synthase inhibitors with the different target subunits of the enzyme on selected bacteria.

ATP Synthase Inhibitor	Targeted Subunit	Bacteria	Reference
Resveratrol	subunit β and c-terminal region of subunit γ	*S. aureus*	[28]
Venturicidin A	subunit c	*E. coli* *P. aeruginosa* *P. denitrificans* MRSA *Enterococcus*	[66,67,68,69]
Bedaquiline	subunit c and ε	*M. tuberculosis* Selective effect against: *Nocardia* *Corynebacterium*, *S. pneumonia* *S. aureus* *E. faecalis* *E. coli*, *H. pylori*, *H. influenza*	[39,40,41,42,43,48]
Tomatidine	subunit c	*S. aureus* *P. aeruginosa* *E. faecalis* *L. monocytogenes*	[54,56,57,59,61]
Piceatannol	Pocket created by α and β stator subunits and the carboxyl-terminal region of the subunit γ rotor	*E. coli* *S. mutans*	[34,35]
Oligomycin A DCCD	subunit c subunit c	ND *E. coli*	[65] [62,63]
**Venoms** Melittin LAAO PLA2 anoplin cupiennin 1a latarcin 1 latarcin 3a latarcin 5 pandinin 2 eumenitin lasiocepsin lycosin1 mastoparanB panurgine1 protonectin cathelicidin BF-30 lycotoxin	subunit β ND βDELSEED-motif βDELSEED-motif βDELSEED-motif βDELSEED-motif	*E. coli* *Pseudomonas* spp. *S. epidermidis* *E. coli* *E. coli* *E. coli* *E. coli*, *P. aeruginosa* *S. aureus* *E. coli*	[77,78,79,80] [81] [79] [82] [83] [83]

LAAO: L-amino-acid-oxidase; PLA2: phospholipase A2; ND: non-determined.

## Data Availability

Not applicable.

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
