# Peer review of "Inhibitors of ATP Synthase as New Antibacterial Candidates"

_antibiotics, 2023, doi:10.3390/antibiotics12040650_

Round 1
Reviewer 1 Report
This review introduces the structure and function of ATPase, and summarizes the inhibitory effects of several ATPase inhibitors on bacterial ATPase and their clinical applications. The overall structure is brief and relatively clear, but the logical structure needs to be strengthened when introducing ATPase inhibitors. It is not a good idea to present several ATPase inhibitors briefly in the article.
1. Title: The original title "ATP synthase, the enzyme of life: a promising therapeutic target against bacterial infections" can only highlight part of the content. It is suggested that the keyword “inhibitor”, which occupies most of the text, be included.
2. The section "Therapeutical applications" focuses on the inhibition principles and applications of several "ATP synthase inhibitors". The narrative appears to be a mere non-sequential presentation of the inhibitors. Thus, it is recommended to present them in a logical order with subheadings. For example, the inhibition of ATPase could be performed according to the main binding subunits, which would echo the introduction of the ATP structure in the first half of the article. Or other logical or even chronological order that the authors consider more reasonable to introduce "ATP synthase inhibitors".
3. Line 151-180 mainly introduces the principle and application of bedaquiline suppression, and the beginning of this paragraph suddenly mentions "According to the world health organization (WHO), tuberculosis (TB) ... ..." is very broken. Until the appearance of "bedaquiline" in line 155, there is no hint that it is "A inhibitor of ATPase", so it is suggested to improve the narration of this paragraph.
4. Line 178-180 mentions the reason why "bedaquiline" is so "interesting" is due to its effect on the immune system in addition to its inhibitory effect on ATPase. This point can be put in the prospective (refer to the 5th revision suggestion) as a major point of investigation for later researchers to explore this field.
5. Line250-265: Two ideas in this paragraph are to genetically modify ATPase in the expectation of reducing ATPase activity, and to develop inhibitors that capture the vast differences between human ATPase and bacterial ATPase. The former is suggested to be described in the outlook section, and the latter as a key entry point for the development of novel ATPase inhibitors.
6. The title of Table 1. should be located on the top of the table.
7. What are the limitations of current ATP inhibitors that should also be reflected in the conclusion section
8. Line 126 has one extra punctuation mark.
Author Response
This review introduces the structure and function of ATPase, and summarizes the inhibitory effects of several ATPase inhibitors on bacterial ATPase and their clinical applications. The overall structure is brief and relatively clear, but the logical structure needs to be strengthened when introducing ATPase inhibitors. It is not a good idea to present several ATPase inhibitors briefly in the article.
- Title: The original title "ATP synthase, the enzyme of life: a promising therapeutic target against bacterial infections" can only highlight part of the content. It is suggested that the keyword "inhibitor", which occupies most of the text, be included.
Done. We changed the title to another that highlights the main idea of our review.
- The section "Therapeutical" application focuses on the inhibition principles and applications of several "ATP synthase inhibitors". The narrative appears to be a more non-sequential presentation of the inhibitors. Thus, it is recommended to present them in a logical order with subheadings. For example, the inhibition of ATPase could be performed according to the main binding subunits, which would echo the introduction of the ATP structure in the first half of the article. Or other logical or even chronological order that the authors consider more reasonable to introduce "ATP synthase inhibitors".
We separated the inhibitors into different paragraphs each entitled by the name of the inhibitor.
- Line 151-180 mainly introduces the principle and application of bedaquiline suppression, and the beginning of this paragraph suddenly mentions "According to the world health organization (WHO), tuberculosis (TB) ... .." is very broken, Until the appearance of "bedaquiline" in line 155, there is no hint that it is "An inhibitor of ATPase", so it is suggested to improve the narration of this paragraph.
Done.
- Line 178-180 mentions the reason why "bedaquiline" is so "interesting" is due to its effect on the immune system in addition to its inhibitory effect on ATPase. This point can be put in the prospective (refer to the 5th revision suggestion) as a major point of investigation for later researchers to explore this field.
Done. We discussed more this point.
- Line250-265: Two ideas in this paragraph are to genetically modify ATPase in the expectation of reducing ATPase activity, and to develop inhibitors that capture the vast differences between human ATPase and bacterial ATPase. The former is suggested to be described in the outlook section, and the latter as a key entry point for the development of novel ATPase inhibitors.
We thank the reviewer for this comment. We have made the changes.
- The title of Table 1 should be located on the ion of the table.
Done
- What are the limitations of current ATP inhibitors that should also be reflected in the conclusion section?
We added the limitations of each inhibitor, if any, at the end of each paragraph and we recaptured it in the conclusion.
- Line 126 has one extra punctuation mark.
Done.
Reviewer 2 Report
Overall, this review article is okay. However, much of the content/info has been reported/discussed in the previously published review article within the last few years. For examples:
10.2174/0929867311320150003;
10.2174/092986710791859270;
doi.org/10.1021/acs.jnatprod.9b01024
The authors must show the readers the uniqueness and significance of their review. In addition, the current references the authors used seem insufficient to become a full review article. The current figures in the review article are some fundamental textbook schematics. The authors need to think about the recent advanced and perspective research in this area and come up with some unique/informative figures and contents (and also an excellent graphical abstract)
Author Response
The authors must show the readers the uniqueness and significance of their review. In addition, the current references the authors used seem insufficient to become a full review article. The current figures in the review article are some fundamental textbook schematics. The authors need to think about the recent advanced and perspective research in this area and come up with some unique/informative figures and contents (and also an excellent graphical abstract).
We thank the reviewer for this comment. When we were working on this review we didn’t find any review that discusses all the types of inhibitors in one review (natural inhibitors like spices, animal venoms, antibiotics and chemical inhibitors). Concerning the references and figures, we have made the changes. We added more references, and a figure that shows the structure of ATP synthase and where the important inhibitors work on. Also we added a graphical abstract.
Reviewer 3 Report
General comments:
The authors review ATP synthase inhibitors as a potential therapeutic strategy against bacterial infections. This manuscript nicely overviews the structure and function of bacterial ATP synthase and various inhibitors. This information would be informative for readers; however, some issues need to be clarified before this manuscript can be accepted.
Specific comments:
1. L. 303, Although the authors claim the difference between bacterial and human ATP synthases, the difference is not described enough in the text. Since, instead, the structural similarity of these synthases is demonstrated, the authors are advised to discuss the difference in detail.
2. L. 304, Although the authors claim that bacterial ATP synthase inhibitors do not affect human health, this sentence seems inappropriate. For instance, Bedaquiline treatment increases the risk of cardiac arrhythmias and death (Commun Biol 3, 452 (2020)). For another example, Venturicidin is reported to be a potent inhibitor of both mitochondrial and bacterial ATP synthase (Eur J Biochem 46, 157–167 (1974), Biochim Biophys Acta 807, 238–244 (1985), Microbiol Mol Biol Rev 72, 590–641 (2008)). Since these data are against the authors’ argument, they need to discuss this point by showing these potential side effects of bacterial ATP synthase inhibitors when used for infected patients.
3. How do the authors think about the emergence of ATP synthase inhibitor-resistant bacteria? If any research to address this point is already present, the authors are advised to discuss this point by referring to those studies since the possibility of the emergence of resistant bacteria is always a concern for patients, clinicians, and microbiologists.
Author Response
The authors review ATP synthase inhibitors as a potential therapeutic strategy against bacterial infections. This manuscript nicely overviews the structure and function of bacterial ATP synthase and various inhibitors. This information would be informative for readers; however, some issues need to be clarified before this manuscript can be accepted.
Specific comments:
1. L. 303, Although the authors claim the difference between bacterial and human ATP synthases, the difference is not described enough in the text. Since, instead, the structural similarity of these synthases is demonstrated, the authors are advised to discuss the difference in detail.
Done. We added the difference in detail.
2. L. 304, Although the authors claim that bacterial ATP synthase inhibitors do not affect human health, this sentence seems inappropriate. For instance, Bedaquiline treatment increases the risk of cardiac arrhythmias and death (Commun Biol 3, 452 (2020)). For another example, Venturicidin is reported to be a potent inhibitor of both mitochondrial and bacterial ATP synthase (Eur J Biochem 46, 157–167 (1974), Biochim Biophys Acta 807, 238–244 (1985), Microbiol Mol Biol Rev 72, 590–641 (2008)). Since these data are against the authors’ argument, they need to discuss this point by showing these potential side effects of bacterial ATP synthase inhibitors when used for infected patients.
We fully agree with the insightful point raised by the reviewer. We added at the end of each inhibitor’s paragraph if there are any adverse effect that should be took into consideration and suggestions (if any) about how to manage these side effects.
3. How do the authors think about the emergence of ATP synthase inhibitor-resistant bacteria? If any research to address this point is already present, the authors are advised to discuss this point by referring to those studies since the possibility of the emergence of resistant bacteria is always a concern for patients, clinicians, and microbiologists.
We thank the reviewer for this comment. We think that there’s a possibility of the emergence of ATP synthase inhibitor-resistant bacteria because as we know bacteria have high mutational ability to adapt. We discussed this issue in the revised version of our manuscript.
Reviewer 4 Report
The topic is interesting. Please revise as follows.
- The reference citation style is not following the MDPI style.
- The organisms names should be written in full at first, then abbreviated name for the rest (text and tables). Please revise.
- Provide a figure for the chemical structures of the most promising compounds acting by inhibiting the ATP synthase.
- In the discussion pat, discuss the limitations and possible drawbacks (if any) for using the natural alternatives.
Author Response
Comments and Suggestions for Authors.
The topic is interesting. Please revise as follows.
- The reference citation style is not following the MDPI style.
Done
- The organisms names should be written in full at first, then abbreviated name for the rest (text and tables). Please revise.
Done
- Provide a figure for the chemical structures of the most promising compounds acting by inhibiting the ATP synthase.
We thank the reviewer for this comment. We added the figures.
- In the discussion part, discuss the limitations and possible drawbacks (if any) for using the natural alternatives.
We have discussed the drawbacks for using natural alternatives at the end of the paragraph entitled “natural spices”. Actually the limitations are dose- related and medication frequency.
Round 2
Reviewer 1 Report
No more comments
Author Response
No more comments
We thank the reviewer and look forward to future cooperation to publish in the Journal Antiobiotcs.

Reviewer 2 Report
It seems that the authors didn't improve the below point as mentioned last time:
- The authors need to think about the recent advanced and perspective research in this area and come up with some unique/informative figures and contents (and also an excellent graphical abstract).
Author Response
It seems that the authors didn't improve the below point as mentioned last time:
The authors need to think about the recent advanced and perspective research in this area and come up with some unique/informative figures and contents (and also an excellent graphical abstract).
Done, see please the revised version of the manuscript. We have added a graphical summary and the necessary recommended information.

Round 3
Reviewer 2 Report
I have no further comments. Thanks!